# Enhancing Magnetic Material Data Analysis with Genetic Algorithm-Optimized Variational Mode Decomposition

Xinlei Jin [1] and Quan Qian [2,3,*]

1   School of Computer Engineering and Science, Shanghai University, Shanghai 200444, China;
    21721681@shu.edu.cn
2   Research Center of Urban Information, Center of Materials Informatics and Data Science, Shanghai University,
    Shanghai 200444, China
3   Key Laboratory of Silicate Cultural Relics Conservation (Shanghai University), Ministry of Education,
    Shanghai 200444, China
*   Correspondence: qqian@shu.edu.cn

**Abstract:** As the application of machine learning technology in predicting and optimizing material performance continues to grow, handling the electromagnetic data of magnetic materials, especially in removing unavoidable data noise and accurately extracting resonance peaks in the imaginary part of electromagnetic information, has become a significant challenge. These steps are crucial for revealing the deep electromagnetic behavior of materials and optimizing their performance. In response to this challenge, this study introduces an innovative approach—Genetic Algorithm-Optimized Variational Mode Decomposition for Signal Enhancement (GAO-VMD-SE). This method, through the Variational Mode Decomposition (VMD) technique optimized by genetic algorithms, not only effectively reduces noise in the data, thereby improving the Signal-to-Noise Ratio (SNR) and reducing the Mean Absolute Error (MAE), but also significantly enhances the hidden resonance peak information in complex permittivity and permeability data to achieve a comprehensive improvement in key performance indicators. Experimental results prove that this method surpasses traditional analysis techniques in key performance metrics such as the peak width ratio, peak overlap ratio, and the number of peaks. Especially in identifying characteristic peaks related to the Snoek limit, GAO-VMD-SE can effectively reveal the peak features hidden in complex data, thus providing important insights for evaluating the performance of materials at specific frequencies. Moreover, the effectiveness of this method in denoising not only enhances the quality and accuracy of material data analysis but also achieves a 1% to 10% enhancement in peak information extraction. This optimized data processing capability and versatility make GAO-VMD-SE not only suitable for evaluating the performance of magnetic materials but also show significant practical application value in processing spectral data and other time series signal data applications.

**Keywords:** variational mode decomposition; data denoising; feature extraction; genetic algorithm

## 1. Introduction

With the rapid development of communication technology, there is a growing demand for materials capable of supporting applications, especially magnetic materials. Due to their key role in enhancing signal transmission efficiency and reducing energy consumption, these materials have become a vital component of modern technological advancements [1,2]. Faced with this challenge, the scientific community is dedicated to surpassing the traditional magnetic materials' Snoek limit to achieve materials of higher performance [3]. The Snoek limit refers to the theoretical maximum value of magnetic anisotropy caused by the lattice distortion in magnetic materials, which has a decisive impact on improving the performance of magnetic materials. Therefore, accurately identifying and characterizing the resonance peaks in magnetic materials—key indicators related to the Snoek limit—is crucial for

transcending the capabilities of traditional materials and fostering the development of high-performance materials for future technologies.

However, in the process of experimental data analysis, noise introduced by experimental equipment, measurement errors, or environmental factors often interferes with the accurate identification of resonance peaks, thereby becoming a significant obstacle to precisely predicting and optimizing material performance. Effectively removing noise, enhancing data quality, and accurately extracting resonance peak information related to the Snoek limit are key to optimizing the performance of magnetic materials [4]. Although past research has proposed various denoising techniques, such as window smoothing [5,6], wavelet transform [7,8], and singular value reconstruction [9–11], thus achieving success in specific scenarios, these methods do not always meet the denoising needs in all scenarios, especially when dealing with magnetic material data.

In response to the above challenges, this study proposes an innovative denoising method—Genetic Algorithm-Optimized Variational Mode Decomposition for Signal Enhancement (GAO-VMD-SE). This method combines the efficiency of Variational Mode Decomposition (VMD) with the optimization capability of genetic algorithms, thereby aiming to effectively remove noise and extract key trends and peak information from material data. In this way, we can not only gain a deeper understanding of the complex characteristics of material data but also comprehensively improve the quality assessment of datasets, thereby providing more accurate and reliable input data for machine learning models. The development of this method aims to provide researchers in the field of materials science with a powerful tool to optimize material data utilization, thus promoting further improvement in material design and performance.

The core objective of this study is to explore a new method for effectively handling noise in material data to improve data quality and optimize the performance of machine learning models. Through the GAO-VMD-SE method, we hope to contribute a new analytical tool and methodology to the field of materials science, thereby aiding in the design and discovery of advanced materials suitable for a wide range of technological applications.

The technical contributions of this study are mainly reflected in the following aspects:

- Genetic Algorithm-Optimized Variational Mode Decomposition (VMD) parameter selection: This paper introduces genetic algorithms to optimize the parameters of VMD, which is a strategy that takes into account multiple variables in the VMD process and seeks the optimal parameter combination through an iterative process. This improvement significantly enhances the efficiency and quality of data processing, especially in dealing with complex and magnetic material data, thereby enabling the more precise identification and extraction of key peak information to adapt to complex data characteristics.

- Application of high-modal Variational Mode Decomposition (VMD): By adopting a high-modal VMD method and setting a central frequency threshold, this study has effectively denoised the data, thereby achieving efficient noise removal. This step ensures the authenticity and reliability of the processed data, thus making the data closer to the actual situation.

- Clustering reconstruction based on central frequency and data distance: For the data after VMD decomposition and filtering, a clustering reconstruction method based on central frequency and data distance was used. This method produced two important data curves: one revealing data trends and the other highlighting peak information. By detrending to extract peak information, it ensures that the extracted peaks are not affected by the overall data trend, thus further refining the results.

- Comprehensive data quality assessment: This method reveals peak information masked by data trends and performs a comprehensive quality assessment of the dataset. This includes a comprehensive evaluation of the noise level, trend quality, and the amount of peak information, thereby significantly improving the accuracy of data analysis. Especially in identifying data peaks related to the Snoek limit, this study has shown significant advantages.

The rest of the paper is structured as follows. In Section 2, we will provide a brief review of the background of currently popular methods and discuss their related shortcomings. Section 3 describes the preliminaries and specific methods proposed to address these limitations. Section 4 presents the experimental results and analysis. Finally, in Sections 5 and 6, we will summarize this study and discuss potential directions for future research.

## 2. Related Work

Variational Mode Decomposition (VMD) technology, as an advanced signal processing method, has been extensively applied in the data processing field. Researchers like Wang S et al. [12] have successfully combined VMD with coherence factor methods to significantly enhance the quality of ultrasound CT (USCT) images and effectively remove noise. In addressing the issue of speckle noise in ultrasound imaging, ref. [13] proposed a frequency division denoising algorithm, integrating transform and spatial domain approaches, which outperformed existing methods in preserving structural details and reducing noise. Similarly, Ohmichi Y [14] utilized VMD to extract and analyze coherent structures from high-dimensional spatiotemporal data, thus gaining empirical validation in the field of fluid dynamics. Li H et al. [15] identified bearing fault types effectively by maximizing envelope kurtosis for modal number determination, which was coupled with an optimal intrinsic mode function selection technique based on band entropy. Moreover, Meng Z et al. [16] proposed an optimized VMD method based on an improved scale space representation, thereby enhancing the accuracy of fault feature extraction in rolling bearings. These studies not only demonstrate the effectiveness of VMD in denoising and enhancing data analysis accuracy but also showcase its potential in mining information from complex data, which is significant for material design and performance optimization.

In the research and application fields of magnetic materials, the importance of these materials has increasingly become prominent over time. Especially in Additive Manufacturing (AM) technology [17], magnetic materials are used to manufacture functional components that play key roles in various applications such as electronic devices, electric motors, and wind turbines. The application of AM technology helps in improving performance while reducing manufacturing costs. Magnetic Nanoparticles (MNPs), as a focus of intelligent material design in the biomedical field [18], exhibit wide application potential in cancer treatment, targeted drug delivery, medical imaging, and biomolecule extraction due to their magnetically controllable properties. For industrial dye pollution, the technique of adsorbing dyes in aqueous solutions using magnetic nanomaterials has become an effective wastewater treatment method [19]. Simultaneously, the study of transverse thermoelectric effects in magnetic materials, especially the anomalous Nernst effect [20], has attracted broad attention in fundamental physics and thermoelectric application fields, thereby heralding new progress in transverse thermoelectric power generation research and offering new ideas for the development of next-generation thermal energy harvesting and thermal flow sensing technologies. These research outcomes indicate the significant technical and application implications of in-depth exploration and performance optimization of magnetic materials.

Although denoising technology plays a key role in improving the quality of material data analysis, such as the work of Liu X et al. [21] enhancing data quality for machine learning applications in material science through denoising and Zhao S et al. [22] addressing the issue of high noise in electromagnetic ultrasonic transducers in high-temperature environments and the difficulty in extracting signal features, existing methods still face many challenges when dealing with complex or high-noise datasets. Especially in magnetic material data analysis, the inherent fluctuation characteristics and complexity make it difficult for existing methods to accurately capture key features, thereby limiting the precise evaluation of magnetic material performance.

In light of these considerations, this paper presents an innovative approach that integrates the strengths of VMD technology with additional denoising techniques, and it is

tailored to tackle the distinct challenges encountered in magnetic material data analysis. Through meticulous optimization of VMD parameter selection and denoising methodologies, this study endeavors to enhance the precision of extracting and analyzing pivotal performance metrics of magnetic materials. Furthermore, this paper delves into enhancing the accuracy and efficiency of data analysis through the utilization of machine learning algorithms, thereby aiming to pave the way for novel research avenues and practical applications within the realm of materials science.

### 3. Preliminaries and Methods

The overall process of the GAO-VMD-SE algorithm framework is illustrated in Figure 1. It primarily consists of three key steps: genetic algorithm-optimized Variational Mode Decomposition (VMD), data denoising and clustering reconstruction, and the extraction of target peak information. Initially, this study employed a genetic algorithm to optimize the key input parameters of VMD, namely the number of modes $K$ and the penalty factor $\alpha$. By minimizing the envelope entropy as the optimization objective function, this method ensures that VMD effectively captures key frequency components during the signal decomposition process, thereby enhancing the precision and reliability of signal processing. It is noteworthy that the precise setting of parameters $K$ and $\alpha$ is crucial for optimizing VMD performance, thereby affecting the balance between signal reconstruction accuracy and the suppression of overfitting.

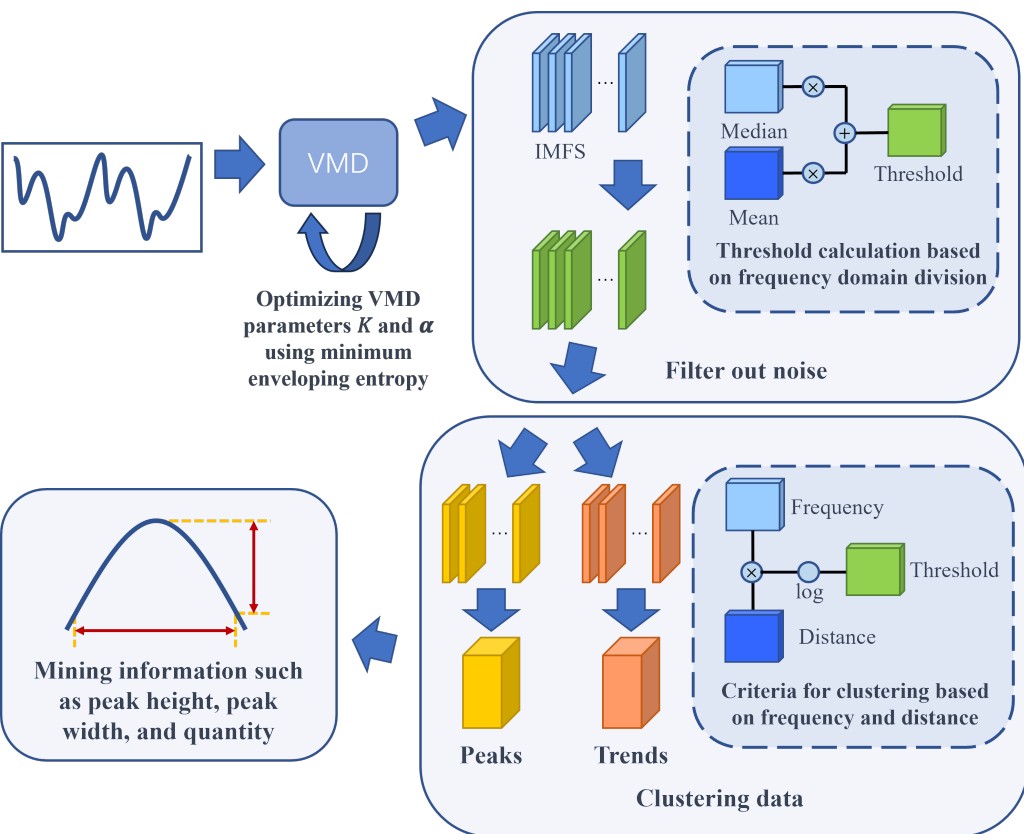

**Figure 1.** The framework of GAO-VMD-SE algorithm.

After obtaining the optimized parameter settings, the algorithm performs VMD decomposition on the data, thus producing multiple Intrinsic Mode Functions (IMFs). At this stage, an adaptive threshold selection method is utilized to remove noise IMFs from the decomposition results. This method takes into account the median and mean of the signal to accurately identify and eliminate noise components.

Subsequently, a clustering analysis of the IMFs is conducted, thereby dividing the remaining IMFs into two major categories: those reflecting data trends and those revealing

data peak features. This clustering process is based on the central frequency of the IMFs and their distance from the original data, thus aiming to provide a more precise data foundation for subsequent analysis. Finally, within the data peak class IMFs, target peak information is identified and extracted through detailed analysis. These key peaks provide rich information for understanding the data, thereby playing a crucial role in subsequent data analysis and application.

### 3.1. Variational Mode Decomposition

Variational Mode Decomposition (VMD) is a signal processing technique widely used for the decomposition and analysis of nonstationary signals. The core idea of VMD is to decompose a signal into $k$ band-limited Intrinsic Mode Functions (IMFs) $u_k(t)$, where each IMF has distinct frequency and amplitude characteristics, thereby representing the local frequency components of the data. The specific steps are as follows:

Firstly, compute the analytic signal of each mode $\mu_k$ through the Hilbert transform, thus obtaining the one-sided spectrum of each mode function through Equation (1)

$$\left( \delta(t) + \frac{j}{\pi t} \right) \times u_k(t) \tag{1}$$

where $\delta(t)$ represents the shock function, $t$ is time, and $u_k(t) = u_1(t), u_2(t), \ldots, u_k(t)$ represents the $k$ modal components obtained through VMD.

Secondly, modulate the center frequency through exponential mixing and convert the spectrum of each mode to the corresponding baseband. The process is shown in Equation (2).

$$\left[ \left( \delta(t) + \frac{j}{\pi t} \right) \times u_k(t) \right] e^{-j\omega_k t} \tag{2}$$

where $\omega_k(t) = \omega_1(t), \omega_2(t), \ldots, \omega_k(t)$ represents the center frequency of each $u_k(t)$.

Thirdly, the demodulated signal's Gaussian smoothing and the squared gradient criterion allow us to estimate the bandwidth of each modal component. Thus, a constrained variational model can be constructed and represented through Equation (3).

$$\begin{cases} \min_{\{u_k k\}, \{\omega_k\}} \left\{ \sum_{k=1}^{K} \left\| \partial_t \left[ \left( \delta(t) + \frac{j}{\pi t} \right) \times u_k(t) \right] e^{-j\omega_k t} \right\|_2^2 \right\} \\ \sum_{k=1}^{K} u_k(t) = f \end{cases} \tag{3}$$

For solving the variational fraction problem, a sufficiently large positive number as the quadratic penalty factor can ensure the reconstruction accuracy of the input signal, while the Lagrange penalty operator can make the reconstruction more precise. Therefore, by introducing the quadratic penalty factor $\alpha$ and the Lagrange penalty operator $\lambda$, the constrained variational problem can be transformed into an unconstrained variational problem. Their combination can be written as a generalized Lagrangian expression through Equation (4).

$$L(\{u_k\}, \{\omega_k\}, \lambda) = \alpha \sum_k \left\| \partial_t \left[ \delta(t) + \frac{j}{\pi t} \times u_k(t) \right] e^{-j\omega_k t} \right\|_2^2 + \left\| f(t) - \sum_i u_i(t) \right\|_2^2$$
$$+ \left\langle \lambda(t), f(t) - \sum_k u_k(t) \right\rangle. \tag{4}$$

To obtain the "saddle point" of the generalized Lagrangian expression, $L$, $\mu_k$, $\omega_k$, and $\lambda$ are iteratively updated through the Alternating Direction Method of Multipliers (ADMM), thereby obtaining the optimal solution of the constrained variational model within the

precision requirements. To update $\mu_k^{n+1}$, the following equation (Equation (5)) can describe the corresponding minimization problem.

$$u_k^{n+1} = \operatorname*{argmin}_{u_k \in x} \left\{ \alpha \left\| \partial_t \left[ \delta(t) + \frac{j}{\pi t} \times u_k(t) \right] e^{-j\omega_k t} \right\|_2^2 + \left\| f(t) - \sum_i u_i(t) + \frac{\lambda(t)}{2} \right\|_2^2 \right\} \quad (5)$$

Finally, by converting the formula to the frequency domain through the Parseval/Plancherel Fourier isometry transform, the minimization problem can be solved using Equation (6).

$$\widehat{u_k^{n+1}}(\omega) = \frac{f(\omega) - \sum_{i \neq k} \widehat{u_i}(\omega) + \frac{\widehat{\lambda(\omega)}}{2}}{1 + 2\alpha(\omega - \omega_k)^2} \quad (6)$$

where $\widehat{u_k^{n+1}}$ represents the current signal Wiener filtering, and $u_k(t)$ is the real part obtained after Fourier transforming $\{\widehat{u_k}(t)\}$.

Following the same steps, the center frequency $\omega_k$ is updated using Equation (7):

$$\omega_k^{n+1} = \frac{\int_0^\infty \omega \left| \widehat{u_k(w)} \right|^2 d\omega}{\int_0^\infty \left| \widehat{u_k(w)} \right|^2 d\omega} \quad (7)$$

For accurate and effective signal reconstruction, the Lagrange multiplier is updated using Equation (8):

$$\widehat{\lambda^{n+1}}(\omega) = \widehat{\lambda^n}(\omega) + \tau \left( \hat{f}(\omega) - \sum_k \widehat{u_k^{n+1}}(\omega) \right) \quad (8)$$

The entire iterative process is modified using Equations (7) and (8) to determine if the constraints are satisfied. If they are, the loop stops to obtain a finite number of IMF components. If not, it returns to Formula (6) and repeats the cycle until the convergence stop condition is met. The termination condition is defined by Equation (9).

$$\sum_k \left\| \hat{u}_k^{n+1} - \hat{u}_k^n \right\|_2^2 / \left\| \hat{u}_k^n \right\| < \varepsilon \quad (9)$$

where $\varepsilon$ is the precision for convergence determination, and $\varepsilon > 0$.

This process demonstrates the importance of effectively setting the VMD parameters during the material data decomposition process to retain all information from the original sequence as much as possible, thereby avoiding modal mixing, which is crucial for the decomposition of material data and the accuracy of the final model prediction. The mode number *K* and quadratic penalty factor $\alpha$ are parameters that need to be preset in VMD decomposition, and these parameters must adaptively change according to different signals to achieve the best decomposition results.

The analysis underscores the critical balance required in selecting the parameters *K* (the number of modes) and $\alpha$ (the bandwidth parameter) for optimal Variational Mode Decomposition (VMD) performance. A key finding is that an inadequately small *K* leads to underdecomposition, meaning that not all components of the data are adequately separated. This results in the loss of crucial information embedded within the original dataset. Conversely, setting *K* too high results in overdecomposition, thereby introducing nonexistent components or erroneously splitting signals of the same frequency into multiple components.

Similarly, the choice of $\alpha$ significantly influences the decomposition outcome. A lower $\alpha$ value enlarges the bandwidth of the modal components, thereby causing an overlap where distinct components might be erroneously grouped together. On the other hand,

an excessively high $\alpha$ narrows the modal components' bandwidth excessively, thereby risking the omission of vital information from the original signal.

Therefore, selecting an appropriate $[K, \alpha]$ pair is imperative for conducting VMD on material data. This choice is crucial to mitigating modal mixing—a common issue in noisy environments—and enhancing the decomposition process's speed and accuracy. Proper parameter selection not only preserves the integrity of the original data but also ensures a more precise and efficient analysis, thereby underscoring the importance of a methodical approach to parameter optimization in VMD applications.

### 3.2. Multiobjective Genetic Algorithm

To solve multiobjective optimization problems, the Multiobjective Genetic Algorithm (MOGA) is utilized. This optimization technique, based on genetic algorithms, effectively handles the simultaneous optimization of multiple objective functions. MOGA maintains a population of individual solutions, generating new solutions through genetic operations such as crossover and mutation, and evaluates the quality of the solutions using a fitness assessment function. Compared to single-objective genetic algorithms, the MOGA introduces the concept of the Pareto front to better capture the trade-offs between multiple objectives. The basic process is as follows:

(1) Initialization: Generate a set of population individuals constituting the initial population. Each individual represents a potential solution to the problem and is composed of the problem's decision variable values.

$$P = X_1, X_2, \ldots, X_N \tag{10}$$

where $P$ is the population, $N$ is the population size, and $X_i$ is the $i$th individual represented as a vector.

(2) Fitness Calculation: Calculate the fitness for each individual, i.e., the objective function values of the problem. Since it is a multiobjective problem, the fitness of each individual is a multidimensional vector.

$$Fitness(X_i) = f_1(X_i), f_2(X_i), \ldots, f_M(X_i) \tag{11}$$

where $f_j(X_i)$ represents the value of individual $X_i$ on the $j$th objective function.

(3) Pareto Dominance Sorting: Sort individuals by comparing their Pareto dominance relationships. Given two solutions $A$ and $B$, $A$ dominates $B$ if

$$\begin{cases} \forall j \in \{1, 2, \ldots, M\} : f_j(A) \leq f_j(B) \\ \exists k \in \{1, 2, \ldots, M\} : f_k(A) < f_k(B) \end{cases} \tag{12}$$

(4) Selection, Crossover, and Mutation: Select individuals from the Pareto front to generate new individuals for the next generation. The crossover operation is used to exchange genetic information between two individuals, and mutation introduces new diversity through minor changes.

(5) Iteration: Repeat the above calculation, selection, crossover, and mutation processes until a predetermined number of iterations or a convergence criterion is reached.

### 3.3. Fitness Function

To ensure that each Intrinsic Mode Function (IMF)'s envelope is smooth and minimizes entropy, thereby better representing the essence of the signal, we use the minimum envelope entropy as the fitness function in the genetic algorithm. The more complex the time series, the greater the calculated value of envelope entropy and vice versa. When the number of decompositions $K$ is small, it may lead to insufficient signal decomposition, thereby mixing other interference items into the trend term and causing the envelope entropy value to increase. When an appropriate $K$ value is taken, the envelope entropy of the trend term

becomes smaller. Therefore, minimizing the smallest entropy among the decomposed IMFs (local envelope entropy) results in the optimal VMD decomposition.

In specific calculations, the signal is initially transformed using the Hilbert transform, which is a mathematical tool commonly employed to extract signal envelopes that is particularly suitable for nonstationary signals. Its formula is presented in Equation (13).

$$H[s(t)](t) = \frac{1}{\pi} \int_{-\infty}^{\infty} \frac{s(\tau)}{t - \tau} d \tag{13}$$

The result of the Hilbert transform is a complex function where the real part represents the original signal, and the imaginary part represents the signal's Hilbert transform. The complex form of the Hilbert transform is defined by Equation (14):

$$s_H(t) = s(t) + iH[s(t)](t) \tag{14}$$

The magnitude of the complex signal obtained through the Hilbert transform, i.e., the absolute value, can represent the envelope of the original signal. The equation for extracting the envelope is represented by Equation (15).

$$h(t) = |s_H(t)| \tag{15}$$

For each IMF, through an iterative process, a series of envelopes is calculated. Assuming that the $i$th IMF is $c_i(t)$, its envelope is $h_i(t)$; then, we have the following:

$$h_i(t) = |c_i(t)| \tag{16}$$

The minimum envelope entropy is determined by minimizing the entropy of each IMF's envelope. For the $i$th IMF, the calculation formula for the envelope entropy can be expressed by Equation (17):

$$E_i = - \sum_{j=1}^{M} p_i(j) log(p_i(j)) \tag{17}$$

Our ultimate goal is to minimize the sum of the envelope entropies of all the IMFs, i.e., $E_{total} = \sum_{i=1}^{M} E_i$, where $M$ is the number of IMFs.

*3.4. Noise Filtering*

To further improve the quality of the data, we employed a histogram-based noise filtering method. This method filters out low-frequency and high-frequency noise components by analyzing the frequency distribution of the signal and selecting an appropriate threshold.

First, we performed a histogram analysis on the signal after VMD decomposition. The histogram represents the distribution of different frequency components in the signal. We calculated and visualized the histogram for each IMF to help us understand the frequency characteristics of the signal.

Based on the analysis results from the histogram, we employed an adaptive threshold selection method to determine which frequency components should be considered as noise. Specifically, we chose an appropriate threshold $f_{threshold}$ in the histogram calculated using Equation (18):

$$f_{threshold} = a \times median + (1 - a) \times mean \tag{18}$$

where *median* represents the median of the histogram, *mean* represents the mean of the histogram, and $\omega_1$ and $\omega_2$ are two adjustable parameters.

According to the selected threshold $f_{threshold}$, components in the signal with frequencies below or above $f_{threshold}$ are identified as noise and filtered out. This can be achieved using Equation (19).

$$\widehat{u_k}(t) = \begin{cases} u_k^{(t)}, if \ f_k \leq f_{threshold} \\ 0, otherwise \end{cases} \tag{19}$$

where $\widehat{u_k}(t)$ represents the $k$th IMF after noise filtering.

By employing this histogram threshold selection and noise filtering method, we can more accurately identify and remove noise, thereby making the data purer and providing a more reliable foundation for subsequent analysis and model training.

### 3.5. Reconstruction Clustering

After noise filtering, the remaining IMFs are clustered into two categories: a data trend class and data peak class. Since using central frequency or correlation analysis alone is not effective in distinguishing the data, a method combining central frequency with the distance from the original data was used to classify the IMFs. The calculation formula is as shown in Equation (20).

$$Score = log(Fre * Dist) \tag{20}$$

For the distance calculation, we utilized the Minkowski distance with $p = 2$, equivalent to the Euclidean distance, to compute the distance between time series data. This method serves as a general approach for measuring the distance between two vectors or data points. The calculation formula is depicted in Equation (21).

$$Dist(A, B) = \sqrt{\sum_{i=1}^{n} (A_i - B_i)^2} \tag{21}$$

The lower the evaluation score of the Intrinsic Mode Functions (IMFs), the lower the frequency and the closer the distance to the original data, which corresponds more closely with the data trend class. Conversely, higher evaluation scores align more with the data peak class.

Next, we introduced a critical threshold denoted as $S_{threshold}$, which serves to divide the IMFs into two distinct classes. Specifically, we delineated two clustering clusters, namely the data trend cluster $C_{trend}$ and the data peak cluster $C_{peak}$. IMFs characterized by evaluation scores surpassing the threshold were allocated to the data trend cluster, whereas those scoring equal to or below the threshold were allocated to the data peak cluster. These allocations are mathematically formalized through the following equations in Equations (22) and (23):

$$C_{trend} = \{k \in X | S_k \leq S_{threshold}\} \tag{22}$$

$$C_{peak} = \{k \in X | S_k > S_{threshold}\} \tag{23}$$

Finally, by recombining the IMFs of the data trend cluster and data peak cluster, we can obtain the trend information $T(t)$ and peak information $P(t)$ of the data. This can be achieved as follows in Equations (24) and (25):

$$T(t) = \sum_{k \in C_{trend}} \widehat{u_k}(t) \tag{24}$$

$$P(t) = \sum_{k \in C_{peak}} \widehat{u_k}(t) \tag{25}$$

where $\widehat{u_k}(t)$ is the $k$th IMF after noise filtering.

The setting of the threshold $S_{threshold}$ uses the same method as referenced in the noise filtering formula. Through this reconstruction clustering method, we can more finely depict the trends and peaks of the data and further extract complete peak information on data peaks, thereby providing more dimensions of data for subsequent machine learning exploration.

### 3.6. Peak Extraction Method

In peak extraction methods, half-peak calculation is a commonly used technique that helps in precisely determining the position of the peak. A half peak refers to the position

where the height of the peak drops to half of the peak value. By calculating the half peak, we can obtain more accurate peak information. Specifically, the steps for half peak calculation are as follows:

(1) Finding peak position: Mark the position of the peak in the data, i.e., the index of the local maximum, which is denoted as $i_{peak}$.

(2) Calculating half-peak height: Let the height of the peak, i.e., the height from the peak to the trough, be $H_{peak}$; then, the half-peak height $H_{half}$ is $\frac{H_{peak}}{2}$.

(3) Half-peak position: Search on both sides of the peak to find the two boundaries $i_{left}$ and $i_{right}$ such that the horizontal line is at the half-peak height $H_{half}$. These two positions are the left and right boundaries of the half peak.

(4) Half-peak width: The width of the half peak can be calculated by $(i_{right} - i_{left})$. This indicator allows for a more refined analysis of the signal features, thereby offering a quantitative measure of the peak width at half of the maximum height, which is particularly useful in applications requiring precise feature extraction from complex signals.

## 4. Experimental Results

### 4.1. Quantitative Metrics

4.1.1. Denoising Evaluation Metrics

To assess the denoising capabilities of various methods on data, we used two metrics: the Signal-to-Noise Ratio (SNR) and Mean Absolute Error (MAE). The SNR is commonly used to compare the intensity of data information against noise, while the MAE can better reflect the actual situation of the noise. They are defined in the following:

(1) Signal-to-Noise Ratio (SNR): The SNR is a key indicator of data quality and is especially important in dealing with noisy signals. The calculation formula is shown in Equation (26).

$$SNR = 10 * log\left(\frac{P_S}{P_N}\right) = 10 * log\left(\frac{\sum_{i=1}^{N} x_i^2}{\sum_{i=1}^{N}(x_i - \widetilde{x}_i)^2}\right) \tag{26}$$

where $x_i$ represents the value in the original data, and $\widetilde{x}_i$ represents the value after noise reduction. $x_i - \widetilde{x}_i$ can be considered as the noise part. The SNR measures the signal quality by comparing the power of the original signal with the power of the noise. A high SNR implies that the signal strength is higher relative to the noise, thereby typically indicating better data quality.

Additionally, the SNR can be leveraged to generate noise for simulating real-world scenarios or enhancing data diversity. The process involves calculating the noise power based on the desired SNR level and then adding the calculated noise to the original signal. The calculation formula is shown in Equation (27).

$$\text{Noisy Signal} = \text{Signal} + \sqrt{\frac{\text{Signal Power}}{\text{SNR}}} \times \text{random normal}(0, 1) \tag{27}$$

Here, the noisy signal is obtained by adding Gaussian noise to the original signal based on the desired SNR level. The noise power is determined by the signal power divided by the SNR. This process ensures that the resulting signal maintains the desired SNR level, thus preserving signal quality while introducing realistic noise characteristics.

(2) Mean Absolute Error (MAE): The MAE is a commonly used metric to measure the accuracy of model predictions. The formula for calculating the *MAE* is as follows in Equation (28).

$$MAE = \frac{1}{m}\sum_{i-1}^{m}|y_i - \widehat{y}_i| \tag{28}$$

where $y_i$ is the actual value, $\widehat{y}_i$ is the predicted value, and $m$ is the total number of data points. The MAE measures the average level of absolute differences between the model predictions and the actual values. A lower MAE value indicates smaller differences be-

tween predictions and actual values, thereby suggesting that the model has better prediction accuracy.

### 4.1.2. Quality Assessment of Data

In the studied material dataset, the feature information, frequency of occurrence, and distribution within a batch of data showed significant consistency and regularity. Thus, each data point was decomposed into two curves: a data trend curve and a data peak information curve. We conducted further quality assessment analysis on these two types of data. They are define as follows:

(1) Quality Assessment of Trend Curves: We used the Pearson correlation coefficient to calculate the correlation of the trend curves between pairs of data within a batch. The Pearson correlation coefficient mainly calculates the covariance of two variables divided by the product of their standard deviations, as shown in Equation (29).

$$\rho_{\{X,\,Y\}} = \frac{cov(X,Y)}{\sigma_X \sigma_Y} = \frac{E\left[\,(X - \mu_X)\,(Y - \mu_Y)\,\right]}{\sigma_X \sigma_Y} \tag{29}$$

where $\mu_X(\mu_Y)$ represents the expectation of $X(Y)$, $E\left[(X - \mu_X)(Y - \mu_Y)\right]$ represents the covariance between $X$ and $Y$, and $\sigma_X(\sigma_Y)$ represents the standard deviation of $X(Y)$.

(2) Quality Assessment of Peaks: The specific information, frequency of occurrence, and distribution situation of peaks within a batch of data maintain a certain regularity. Therefore, the quality assessment of peaks adopts the following three criteria: Peak Width Ratio (PWR), Peak Overlap Ratio (POR), and Number of Peaks (NP).

- Peak Width Ratio (PWR): Within a data point, the peak width is delimited by the half-peak height, and the ratio of all points included in the peak to all points in the data is extracted in this way. The calculation formula is shown in Equation (30).

$$PWR = \left(\frac{N_{peak}}{N_{total}}\right) * 100\% \tag{30}$$

- Peak Overlap Ratio (POR): Assess the peak overlap ratio between pairs of data within a batch, i.e., how many points between two data points are calculated within the peak width range. The calculation formula is shown in Equation (31).

$$POR = \left(\frac{N_{overlap}}{\min\left(Width_A,\ Width_B\right)}\right) * 100\% \tag{31}$$

- Number of Peaks (NP): Assess the number of peaks extracted from each data point. The calculation formula is shown as Equation (32).

$$NP = \sum N_{peak} \tag{32}$$

To consider these three indicators comprehensively, a Composite Score (CS) is used to assess the peak quality in the dataset. Each assessment indicator can be assigned a weight based on its importance in specific applications. The calculation formula is shown as Equation (33).

$$CS = w_1 * PWR + w_2 * POR + w_3 * NP \tag{33}$$

(3) Assessment of Peaks Corresponding to the Snoek Limit: To determine whether a peak corresponds to the characteristic peak of the Snoek limit, two factors need to be considered: peak size and frequency position. The size of the peak is compared using the product of the peak height and peak width to approximate its area, while the frequency is expected to be in a higher position. The specific evaluation function is shown in Equation (34).

$$S_i = \omega_1 * f_i + \omega_2 * (h_i * w_i) \tag{34}$$

where $\omega_1$ and $\omega_2$ represent the weights of frequency and peak area, respectively, and $f_i$, $h_i$, and $w_i$ represent the center frequency, peak height, and peak width of the $i$th peak, respectively.

### 4.2. Dataset, Experimental Setup, and Comparative Models

The dataset utilized in this research is derived from data presented in various scientific publications [23–26]. Table 1 describes the dataset. It comprises 201 samples, with each consisting of 400 data points evenly distributed across the frequency range of 2 to 18 GHz. In this study, the independent variables include the material's composition and fabrication process, while the dependent variables are the material's electromagnetic parameters, specifically the complex permittivity and complex permeability. The primary objective of this project is to mine diverse information that can represent the material's characteristics by analyzing its electromagnetic parameters. The detailed information about this dataset underscores the importance of data obtained from precise laboratory measurements and highlights the possibility of gaining a deep understanding of material performance through proper analysis of these data.

**Table 1.** Dataset information.

| Feature | Description |
| --- | --- |
| Sample size | 201 |
| Sampling range | 2–18 GHz |
| Number of data points per sample | 400 |
| Independent variables | Material composition, fabrication process |
| Dependent variables | Electromagnetic parameters (complex permittivity, complex permeability) |
| Objective | To extract diverse information from the electromagnetic parameters of a certain material. |

The experiments were conducted on a system equipped with an AMD Ryzen 5 5600H processor, 16.0 GB of RAM, and running a Windows 11 operating system. The graphics processing was supported by a NVIDIA GeForce GTX 3050ti with 4 GB of dedicated memory. Python 3.7 was utilized as the programming language, with PyTorch version 1.11.0 employed for deep learning tasks. The versions of NumPy and Pandas used in the experimentation environment correspond to NumPy 1.20.3 and Pandas 1.3.3, respectively.

In this study, we have adopted a commonly used partition ratio of 70:30, thus allocating 70% of the data for training purposes and the remaining 30% for testing. Within the dataset, we observed that the number of peaks in the data generally fell within the range of 5 to 20. Based on this observation, we set the noise frequency threshold to not less than 0.1, meaning that the range of a single fluctuation in the data should not be less than 10 sampling points. For noise filtering, the parameters $K$ and $\alpha$ were set to the results optimized by the genetic algorithm, with $K = 20$ and $\alpha = 289$. In the peak quality assessment, we used average weights, meaning that the weight proportion of the three indicators was 0.33 each. In the assessment of characteristic peaks corresponding to the Snoek limit, $\omega_1$ was set to 0.95 and $\omega_2$ to 0.05.

In the denoising experiment, we primarily used two common denoising methods as baselines for comparative analysis to assess the performance of our proposed high-modal VMD decomposition method in processing the material data. These two baseline methods are smoothing denoising and wavelet denoising.

### 4.3. Experimental Results and Analysis

#### 4.3.1. Denoising Experiment on Toy Dataset

Here, we used two toy datasets generated by two given functions, $f_1(x) = sin(x) \cdot x$ and $f_2(x) = 10sin(\frac{x}{2}) + 2sin(3x)$, to conduct denoising experiments. The goal was to

evaluate the performance of various denoising algorithms after adding known Signal-to-Noise Ratio (SNR) noise to clean data. The objective was to verify the data restoration capabilities of these denoising algorithms while comparing their performance in terms of Mean Absolute Error (MAE) and SNR improvement. The denoising effect is depicted in Figure 2, and the detailed result is shown in Table 2.

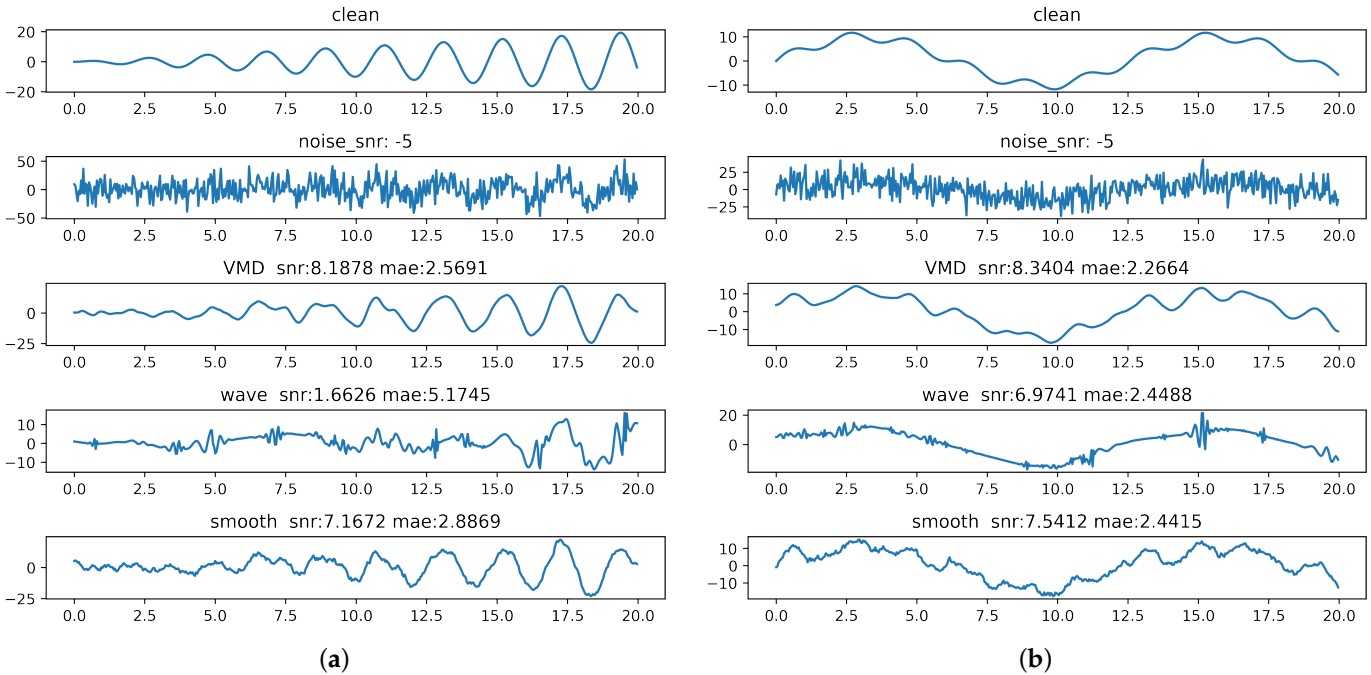

**Figure 2.** The denoising effect diagrams of each model after adding noise to (**a**) $f_1(x)$ and (**b**) $f_2(x)$.

**Table 2.** The denoising effect of each model on the generated data after adding noise. Here, $f_1(x) = sin(x) \cdot x$, and $f_2(x) = 10sin(\frac{x}{2}) + 2sin(3x)$.

| Different Denoising | $f_1(x)$ | | $f_2(x)$ | |
|---|---|---|---|---|
| **Methods** | **SNR** | **MAE** | **SNR** | **MAE** |
| Smoothing Filtering | 7.1672 | 2.8869 | 7.5412 | 2.4415 |
| Wavelet Filtering | 1.6626 | 5.1745 | 6.9741 | 2.4488 |
| VMD | 8.1878 | 2.5691 | 8.3404 | 2.2664 |

According to the results, our method surpassed other models in the denoising effect across multiple generated datasets, thereby achieving a higher SNR and reducing the MAE. This indicates that our method significantly reduced noise, thereby matching closely with the initially added noise and resulting in a smaller MAE for the denoised data compared to the original data. Moreover, the processed data appeared smoother and closer to the original data's waveform characteristics than those treated with other methods.

### 4.3.2. Comparison of Peak Information with Random Parameter Settings

To validate the effectiveness of the genetic optimization algorithm in enhancing the extraction of specific peak information in comparison to random parameter selection, we defined two random parameter combinations, as shown in Table 3. The peak information metric considers peak height, width, and area. And the detailed results are shown in Table 4.

**Table 3.** Random parameters $K$ and $\alpha$ generated by random function.

| Parameter Combinations | $K$ | $\alpha$ |
|---|---|---|
| Random Parameters 1 | 11 | 301 |
| Random Parameters 2 | 8 | 1688 |

**Table 4.** Table of peak information extraction effectiveness using random parameters and optimized parameters. Here, PWR: Peak Width Ratio; POR: Peak Overlap Ratio; NP: Number of Peaks.

| Parameter Settings | PWR | | POR | | NP | |
|---|---|---|---|---|---|---|
| | $e''$ | $u''$ | $e''$ | $u''$ | $e''$ | $u''$ |
| Random Parameters 1 | 10.547 | 13.753 | 0.268 | 0.335 | 5.2532 | 8.6513 |
| Random Parameters 2 | 11.123 | 13.754 | 0.275 | 0.328 | 5.2863 | 8.6217 |
| Optimized Parameters | 11.908 | 14.207 | 0.298 | 0.349 | 5.3125 | 8.8125 |

The results from Table 4, covering changes in the peak width ratio, peak overlap ratio, and the number of peaks under different parameter settings for both $e''$ and $u''$, showed significant improvements. The genetic optimization algorithm notably excelled in optimizing the peak information extraction, thereby demonstrating superior performance in the peak width ratio, overlap ratio, and number of peaks compared to randomly chosen parameters, especially in terms of the peak overlap ratio and number of peaks, thereby highlighting its effectiveness in extracting complex peak information.

4.3.3. High-Modal Decomposition Reconstruction Experiment

To assess and compare the extraction of more peak information after high-modal VMD decomposition reconstruction, two filtering methods were contrasted: smoothing filtering and wavelet filtering. The experiment aimed to evaluate the extraction efficiency and accuracy of each technique by comparing the peak width ratio, peak overlap ratio, and the number of peaks in the processed data. The denosing effect is depicted in Figure 3, and the detailed results are shown in Table 5.

**Table 5.** Table of peak information extracted after filtering with various models: SF (smoothing filtering), WF (wavelet filtering), HM-VMD (high-modal VMD decomposition reconstruction).

| Different Denoising Methods | PWR | | POR | | NP | |
|---|---|---|---|---|---|---|
| | $e''$ | $u''$ | $e''$ | $u''$ | $e''$ | $u''$ |
| SF | 10.728 | 5.533 | 0.280 | 0.292 | 5.25 | 8.3125 |
| WF | 10.508 | 5.405 | 0.277 | 0.298 | 4.8125 | 8.0 |
| HM-VMD | 11.908 | 14.207 | 0.298 | 0.349 | 5.3125 | 8.8125 |

The high-modal VMD reconstruction method effectively separated peak data from the original data, thus enriching the peak information. The decomposition effect was evident, especially in complex permeability data, where smoothing or wavelet processing extracted only two minor peaks; in contrast, VMD high-modal decomposition reconstructed some protrusions in the data's downtrend into peaks, thus increasing the quantity of peak information. Our method's extracted peak information was more complete, thus enhancing the peak width ratio by 1% in complex permittivity and nearly 10% in complex permeability, with an improvement in the peak overlap ratio within a batch of data that confirms the consistent overall trend of peaks in a batch of material data.

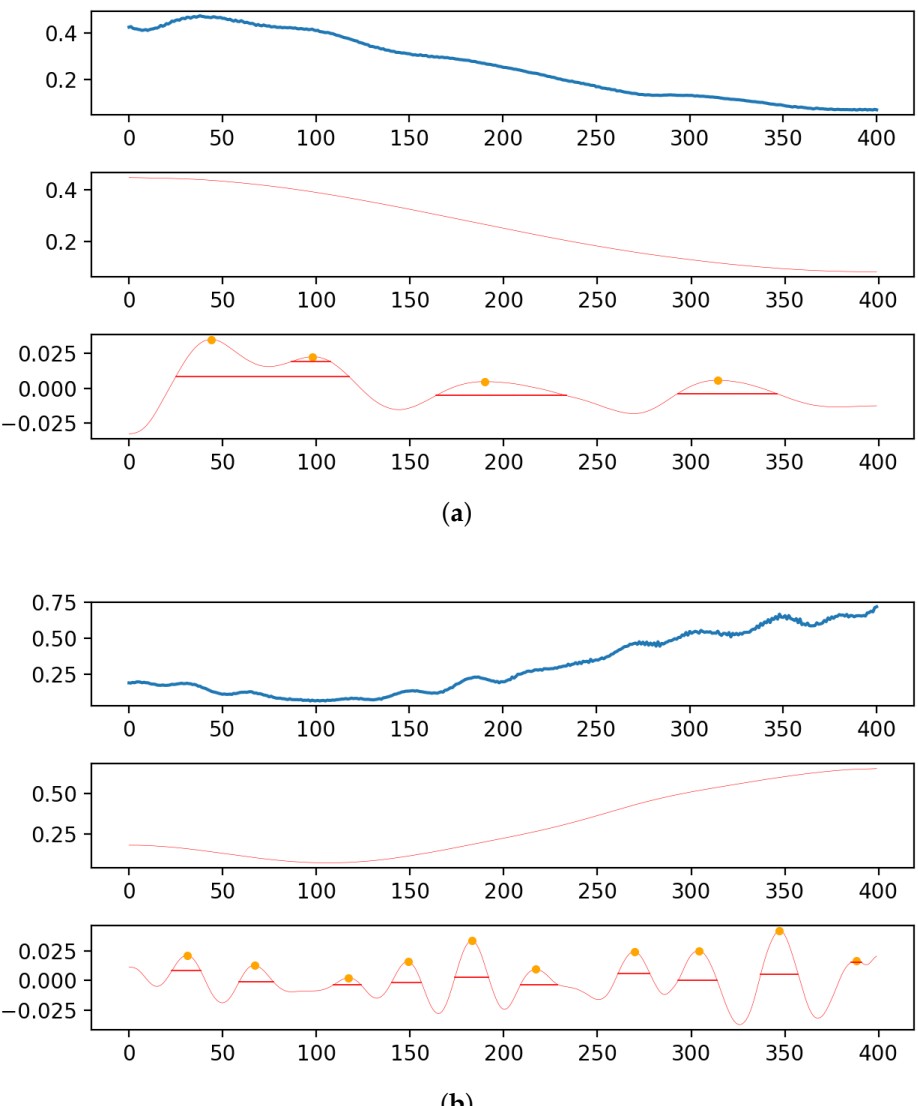

**Figure 3.** Effect diagrams (**a**,**b**) after high-modal VMD decomposition reconstruction.

### 4.3.4. Feature Extraction Regarding the Snoek Limit

Focusing on analysing peak information to identify characteristic peaks corresponding to the Snoek limit, the study concentrated on significant peaks that might represent the Snoek limit through an in-depth analysis of peak information. The final extracted peaks are shown in Figure 4.

This approach evaluated peak height, width, frequency, and relative position to surrounding peaks to identify the most representative peaks. The analysis highlighted the importance of this characteristic peak and the necessity for further in-depth studies to confirm its effectiveness and accuracy as indicative of the Snoek limit, thereby uncovering not readily apparent peak information in the original data and highlighting peaks with clear height advantages and frequency positions suggesting a possible relation to the Snoek limit.

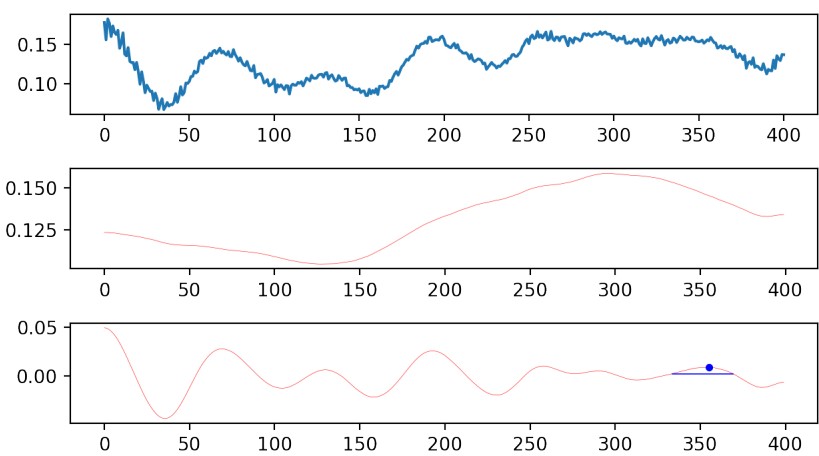

**Figure 4.** Effect diagram of extracting characteristic peaks corresponding to the Snoek limit.

### 4.3.5. Robustness Verification Experiment

To rigorously evaluate the robustness of the GAO-VMD-SE method, we designed an experiment that introduced different degrees of noise into the dataset. This approach was intended to mimic variations in material properties and measurement conditions, thus offering a realistic assessment of the method's denoising capabilities across diverse scenarios. The results of this denoising effect are illustrated in Figure 5.

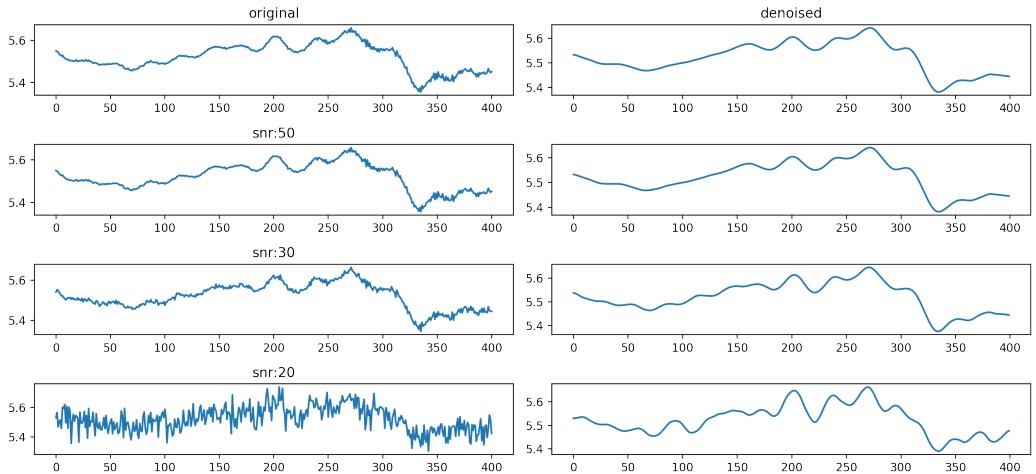

**Figure 5.** Denoising effect after adding noise with different signal-to-noise ratios to the dataset.

This experiment aimed to scrutinize the model's denoising effectiveness under a spectrum of conditions. The GAO-VMD-SE method maintained robust performance when faced with moderate levels of noise, thereby demonstrating its resilience and adaptability to a range of material properties and measurement circumstances. Nevertheless, the method's performance encountered limitations under scenarios characterized by extremely high noise levels. These challenges were largely due to the significant overshadowing of the original signal data by noise, which impeded effective signal extraction and analysis.

In instances of excessive noise ($SNR < 30$), the method's diminished effectiveness highlighted the critical balance between noise levels and signal processing capabilities. Consequently, while the GAO-VMD-SE method proved to be resilient and adaptable within certain noise thresholds, its efficacy was compromised in environments with overwhelming noise interference. This finding emphasizes the necessity of considering the extent of noise and its potential effects on signal processing outcomes when applying the GAO-VMD-SE method in real-world scenarios. Through these observations, we underscore the

importance of adapting and possibly enhancing the method for extreme conditions, thereby ensuring broad applicability and reliability across various material analysis and signal processing applications.

### 4.3.6. Processing Efficiency Analysis

To comprehensively evaluate the scalability of our proposed method for handling large datasets and facilitating real-time analysis applications, we conducted a detailed examination of the processing time required for each data entry, as well as the total runtime across different experimental setups. Given the constraints of our computational resources, which included a CPU with only four cores, we implemented parallel computing techniques utilizing four threads to optimize our processing capabilities. The runtime results are shown in Figure 6.

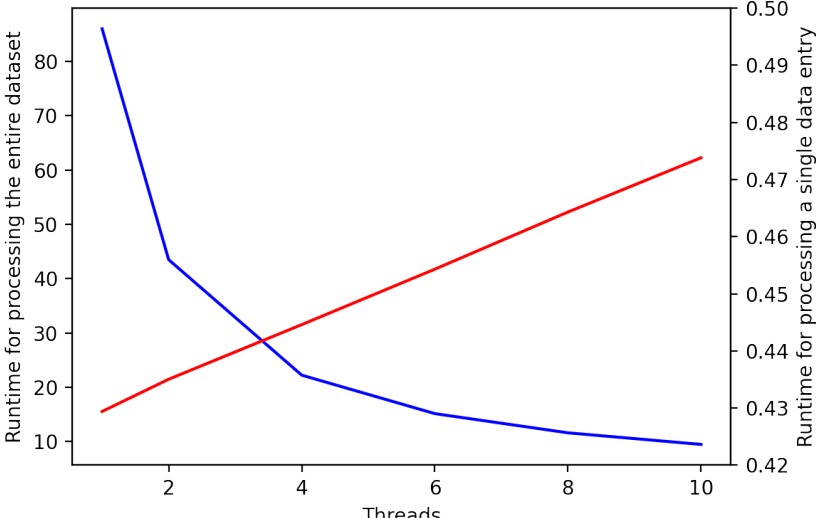

**Figure 6.** Comparison of runtime for processing the entire dataset and a single data entry with varying thread configurations. The plot illustrates the runtime performance achieved by processing the entire dataset (blue line) and a single data entry (red line) across different numbers of threads.

The experimental outcomes clearly indicate that implementing parallel computing strategies can significantly reduce processing times. Specifically, the duration for parallel processing was approximately a quarter of the time required for sequential processing of the entire dataset. This finding underscores our method's capacity to leverage parallel computing effectively, thereby allowing for each data entry to be processed efficiently in parallel configurations.

Consequently, our results confidently support the assertion that our method possesses high scalability, thereby making it ideally suited for processing large datasets and for deployment in real-time analysis scenarios. The integration of parallel computing not only enhances the processing efficiency but also ensures that our method can be effectively applied in demanding real-world contexts, where rapid and resource-efficient data analysis is critical. Through this exploration, we have demonstrated the practical viability of our approach, thus suggesting its broad applicability and potential impact across various domains requiring scalable data processing solutions.

## 5. Discussions and Future Work

This study introduced the Genetic Algorithm-Optimized Variational Mode Decomposition for Signal Enhancement (GAO-VMD-SE) technique, thus effectively enhancing the efficiency and accuracy of electromagnetic data analysis for magnetic materials. While GAO-VMD-SE demonstrates clear advantages in extracting peak information and in denoising applications, its potential extends beyond the specific context of magnetic materials.

The method's fundamental design, focused on denoising signals and extracting vital information from a frequency domain perspective, suggests its applicability could be generalized to various materials and signal types.

Despite the GAO-VMD-SE technique's significant advantages in analyzing the electromagnetic data of magnetic materials, there are still some limitations and challenges. Future research could explore the following directions:

- Further exploration of optimization methods: Although the genetic algorithm was effective in this study, exploring other optimization algorithms could offer additional benefits. For instance, Particle Swarm Optimization (PSO) [27], Simulated Annealing (SA) [28], and deep learning algorithms like Reinforcement Learning (RL) [29] might provide different perspectives on parameter optimization, thus potentially further improving the efficiency and accuracy of VMD. Such comparative studies could also shed light on the most effective optimization strategies for different types of electromagnetic data, thereby enhancing the robustness and adaptability of the denoising and enhancement processes.
- Crossdomain application validation: Verifying the application potential of the GAO-VMD-SE method in other domains, such as biomedical signal processing [30], image processing [13,14], XRD analysis [31] or geophysical data analysis, could expand its application scope.
- Analysis of composite material data: Exploring the effectiveness of the GAO-VMD-SE method in analyzing the electromagnetic data of composite materials or materials with more complex structures to enhance its applicability in new material research.
- Integration with machine learning models: Researching how to integrate the GAO-VMD-SE method with advanced machine learning models to achieve a more automated and intelligent data analysis process.

## 6. Conclusions

This study tackled the challenge of extracting peak information from complex magnetic material electromagnetic data, thereby demonstrating the efficacy of the high-modal Variational Mode Decomposition (VMD) method. Our findings underscore the substantial advantages of employing the VMD approach, particularly in comparison to traditional smoothing and wavelet filtering techniques. The key conclusions include the following:

- Enhanced Peak Information Extraction: The high-modal VMD decomposition reconstruction method significantly outperformed traditional methods in terms of the peak width ratio, peak overlap ratio, and the number of identifiable peaks. This is pivotal for understanding material performance at specific frequencies, especially for revealing characteristic peaks associated with the Snoek limit.
- Improved Data Analysis Quality: The comparison of various denoising techniques underscored the critical role of denoising in augmenting the quality and accuracy of data analysis. Optimizing the VMD parameters through genetic algorithms has markedly increased the decomposition and reconstruction accuracy and efficiency.
- Advancement in Material Performance Evaluation: The GAO-VMD-SE method offers valuable insights and tools for the analysis and enhancement of complex magnetic material data analysis. This contributes to the precise prediction and optimization of material applications, thereby offering robust support for advanced material science research and development.

These findings provide a foundation for further exploration and optimization in the field of magnetic material applications, thereby showcasing the potential for significant advancements in material science.

**Author Contributions:** X.J.: Data curation, Software, Implementation, Investigation, Formal analysis, Visualization, Writing—Original draft. Q.Q.: Conceptualization, Methodology, Funding acquisition, Project administration, Supervision, Writing review and editing. All authors have read and agreed to the published version of the manuscript.

**Funding:** This work was sponsored by the National Key Research and Development Program of China (No. 2023YFB4606200), the Key Program of Science and Technology of Yunnan Province (No. 202302AB080020), and the Key Project of Shanghai Zhangjiang National Independent Innovation Demonstration Zone (No. ZJ2021-ZD-006).

**Data Availability Statement:** The source code that support the findings are available from the corresponding author upon reasonable request.

**Acknowledgments:** The authors gratefully appreciate the anonymous reviewers for their valuable comments.

**Conflicts of Interest:** The authors declare that they have no known competing financial interests or personal relationships that could have appeared to influence the work reported in this paper.

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
