# Peer review of "Enhancing Magnetic Material Data Analysis with Genetic Algorithm-Optimized Variational Mode Decomposition"

_electronics, doi:10.3390/electronics13081408_

Round 1

Reviewer 1 Report

Comments and Suggestions for Authors

This work is interesting and has important potential consequences. But the manuscript needs the following modifications:

1.      Page 4: Authors are recommended to reduce the caption of figure (1) to “The framework of GAO-VMD-SE algorithm.”, since the description of the proposed framework presented in the following text in this section.

2.      Page 9, Lines 317-321: Authors are recommended to revise this paragraph and the related equations (i.e., Eqs. 22 and 23).

3.      It is recommended that the conclusion has one to three sentences described the problem and the conclusion’s main points be presented in bullets.

Author Response

Thank you for your valuable comments and the point-to-point reponse please refer the seperate file. 

Reviewer 2 Report

Comments and Suggestions for Authors

The manuscript introduces a novel method, Genetic Algorithm Optimized Variational Mode Decomposition for Signal Enhancement (GAO-VMD-SE), to enhance the analysis of high-frequency magnetic materials. This approach optimizes Variational Mode Decomposition (VMD) parameters using genetic algorithms, improving data denoising and peak information extraction. The method demonstrates improved performance over traditional techniques across various metrics, providing valuable insights for high-frequency material applications. However, some points remain unclear to me. I would appreciate further clarification from the authors before recommending publication in Electronics.

1. Is it possible that this method can be applied to images, allowing readers to see the intuitive improvements?

2. How were the initial population and the genetic algorithm's hyperparameters, such as the mutation rate and crossover rate, chosen for optimizing VMD parameters? Were alternative optimization techniques considered or compared to assess the genetic algorithm's efficacy?

3. How robust is the GAO-VMD-SE method to variations in material properties and measurement conditions? Were there instances where the method underperformed or failed, and if so, what were the contributing factors?

4. How scalable is the proposed method for processing large datasets or for real-time analysis applications?

5. How generalizable is the GAO-VMD-SE method to other types of materials or signals beyond high-frequency magnetic materials, and are there limitations to its applicability that should be noted?

6. In line 246, is there a typo in that fi(Xi) does not include 'j'?

Comments on the Quality of English Language

I haven't found any further English errors in the manuscript.

Author Response

(The authors gave the same response as above.)

Reviewer 3 Report

Comments and Suggestions for Authors

General Comments: The subject addressed in this article, "Enhancing High-Frequency Magnetic Material Analysis with Genetic Algorithm Optimized Variational Mode Decomposition" is worthy of investigation. The authors propose a method to deal with the problem of noise and peak detection in imaginary part of electromagnetic information in data of high-frequency magnetic materials.

Strengths:

• A good methodology was used by the authors to deal with the problem of noise and peak detection in electromagnetic data. • A good revision of the state-of-the-art was done. • The description of the genetic algorithm for optimizing the variational mode decomposition for Signal Enhancement was well presented.

Weaknesses:

• The authors must extend the discussion about setting parameters of K and α. • There is no information about a real dataset. The authors generate synthetically information to show the performance of the methodology. However, there is not experimental results over real electromagnetic materials.

Author Response

(The authors gave the same response as above.)

Round 2

Reviewer 2 Report

Comments and Suggestions for Authors

The authors addressed all my concerns.

Reviewer 3 Report

Comments and Suggestions for Authors

General comments: The authors have implemented substantial changes in response to the reviewers' feedback, addressing all suggestions. Notably, they emphasized the contribution of their work. They provided detailed information on the parameters K (number of modes) and α (bandwidth parameter) for achieving optimal performance in Variational Mode Decomposition (VMD), as outlined in section 3.1. Moreover, significant enhancements were made to the experimental section, resulting in a more thorough discussion of the results. Based on these improvements, I strongly recommend the paper for publication in its current form.